# Preclinical Validation of an Advanced Therapy Medicinal Product Based on Cytotoxic T Lymphocytes Specific for Mutated Nucleophosmin (NPM1^mut^) for the Treatment of NPM1^mut^-Acute Myeloid Leukemia

**DOI:** 10.3390/cancers15102731

**Published:** 2023-05-12

**Authors:** Marica De Cicco, Ivana Lagreca, Sabrina Basso, Patrizia Barozzi, Stella Muscianisi, Alba Bianco, Giovanni Riva, Sara Di Vincenzo, Chiara Pulvirenti, Davide Sapuppo, Mariangela Siciliano, Vittorio Rosti, Anna Candoni, Marco Zecca, Fabio Forghieri, Mario Luppi, Patrizia Comoli

**Affiliations:** 1SSD Cell Factory e Center for Advanced Therapies, Department of Woman and Child Health, Fondazione IRCCS Policlinico San Matteo, 27100 Pavia, Italy; m.decicco@smatteo.pv.it (M.D.C.); s.basso@smatteo.pv.it (S.B.); s.muscianisi@smatteo.pv.it (S.M.); a.bianco@smatteo.pv.it (A.B.); sara.divincenzo01@universitadipavia.it (S.D.V.); c.pulvirenti@smatteo.pv.it (C.P.); d.sapuppo@smatteo.pv.it (D.S.); m.siciliano@smatteo.pv.it (M.S.); 2Section of Hematology, Department of Medical and Surgical Sciences, University of Modena and Reggio Emilia, AOU Modena, 41124 Modena, Italy; ivana.lagreca@unimore.it (I.L.); patrizia.barozzi@unimore.it (P.B.); acandoni@unimore.it (A.C.); fabio.forghieri@unimore.it (F.F.); mluppi@unimore.it (M.L.); 3SC Pediatric Hematology/Oncology, Department of Woman and Child Health, Fondazione IRCCS Policlinico San Matteo, 27100 Pavia, Italy; m.zecca@smatteo.pv.it; 4Department of Laboratory Medicine and Pathology, Unità Sanitaria Locale, 41126 Modena, Italy; g.riva@ausl.mo.it; 5Center for the Study of Myelofibrosis, General Medicine 2, Fondazione IRCCS Policlinico San Matteo, 27100 Pavia, Italy; v.rosti@smatteo.pv.it

**Keywords:** acute myeloid leukemia, NPM1 mutation, T cell therapy, hematopoietic stem cell transplantation, minimal residual disease

## Abstract

**Simple Summary:**

Nearly 30% of adult acute myeloid leukemias (AML) harbor mutations of the *nucleophosmin* (*NPM1*) gene. These forms have a favorable outcome but, despite notable treatment advances, about 50% of patients still die of progressive disease. Thus, identification of new therapeutic opportunities is important to improve the prognosis. The aim of our study was to assess the feasibility of obtaining a cell therapy medicinal product specific for the mutated NPM1 protein from patients or healthy donors that could be employed to control leukemia and prevent hematologic relapse. We demonstrated that cytotoxic T cells specific for the mutated antigen can be reproducibly expanded, and these cells efficiently recognize and lyse leukemia blasts or other cell types carrying the NPM1-mutated antigen, without causing damage to normal hematopoietic cells. We believe that these T cells may integrate other therapy options in the treatment of patients with refractory or relapsed AML.

**Abstract:**

Acute myeloid leukemia (AML) with nucleophosmin (*NPM1*) genetic mutations is the most common subtype in adult patients. Refractory or relapsed disease in unfit patients or after allogeneic hematopoietic stem cell transplantation (allo-HSCT) has a poor prognosis. NPM1-mutated protein, stably expressed on tumor cells but not on normal tissues, may serve as an ideal target for *NPM1*-mutated AML immunotherapy. The study aim was to investigate the feasibility of producing mutated-NPM1-specific cytotoxic T cells (CTLs) suitable for somatic cell therapy to prevent or treat hematologic relapse in patients with *NPM1*-mutated AML. T cells were expanded or primed from patient or donor peripheral blood mononuclear cells by NPM1-mutated protein-derived peptides, and tested for leukemia antigen-targeted cytotoxic activity, cytokine production and hematopoietic precursor inhibitory effect. We found that mutated-NPM1-specific CTLs, displaying specific cytokine production and high-level cytotoxicity against patients’ leukemia blasts, and limited inhibitory activity in clonogenic assays, could be obtained from both patients and donors. The polyfunctional mutated-NPM1-specific CTLs included both CD8+ and CD4+ T cells endowed with strong lytic capacity. Our results suggest that mutated-NPM1-targeted CTLs may be a useful therapeutic option to control low-tumor burden relapse following conventional chemotherapy in older *NPM1*-mutated AML patients or eradicate persistent MRD after HSCT.

## 1. Introduction

The outcome of acute myeloid leukemia (AML), an heterogeneous hematologic malignancy characterized by the clonal expansion of myeloid blasts, has improved over recent decades, especially in children and young adults who can tolerate intensified treatment strategies, including hematopoietic stem cell transplantation (HSCT) [1,2]. However, relapse is still the major cause of treatment failure in patients who undergo standard induction and consolidation chemotherapy and HSCT [1,2]. Moreover, the management of elderly AML patients, often presenting with comorbidities, remains a major clinical challenge, although in the past few years a high rate of complete remissions were observed in this cohort with the use of low-toxicity agents, such as combined bcl-2 inhibitor venetoclax and hypomethylating agent azacytidine [1,2]. In recent years, the association between the presence of minimal residual disease (MRD) at the end of consolidation therapy and development of hematologic relapse has pointed to the importance of obtaining MRD clearance [1,3]. Hence, novel, low-toxicity strategies to improve post-remission management of AML are being tested, and among those, cell therapy may have a role in complementing existing therapies.

Immunological interventions, such as the use of leukemia-targeted monoclonal antibodies that engage patients’ lymphocytes, or transfer of cancer-specific immune cells, are promising approaches to overcome leukemia resistance to chemotherapy and induce durable remissions [4,5]. Cellular therapies, including T lymphocytes genetically modified to express chimeric antigen receptors (CAR-T), have shown high efficacy in the control of B cell hematologic malignancies [6,7], but major challenges remain to be overcome to limit toxicity and to safely apply CAR-T cell therapy to patients with acute myeloid leukemia [8]. T cell therapies that do not involve genetic modification, such as leukemia antigen-specific cytotoxic T lymphocytes (CTLs), have been developed for AML [9,10,11]. This approach may have the advantage of being more physiologic than CAR-T cell therapy, and thus potentially endowed with fewer side effects, although its efficacy remains to be assessed in controlled clinical trials.

Ideally, AML-specific neoantigens, selectively expressed on malignant myeloid cells and not on normal tissue, would guarantee lower toxicity. *Nucleophosmin* (*NPM1*) mutations are among the most common recurring genetic abnormalities seen in AML, accounting for 30–35% of adult cases [12,13]. These mutations are usually expressed in the entire leukemic population and result in structural changes of the C-terminal of the NPM1 protein, leading to aberrant cytoplasmic delocalization [14]. Mutated NPM1 protein (NPM1^mut^) may be considered a neoantigen, and it is able to elicit a T cell response that has been found to contribute to the maintenance of long-lasting complete remission (CR) in patients treated with conventional chemotherapy [15,16]. Thence, an NPM1^mut^-targeted treatment, based on the use of a patient-derived somatic advanced therapy medicinal product (ATMP) delivered upon the finding of MRD or hematologic relapse after conventional or low-toxicity therapy, may help promote (re)induction of remission in patients with *NPM1*-mutated AML not amenable to HSCT, and contribute to obtain pre-transplant MRD negativity in candidates for allo-HSCT [17]. Moreover, NPM1^mut^-specific T cells may be expanded from HSCT donors, and used in a targeted donor lymphocyte infusion (DLI) protocol to control hematologic relapse after HSCT [17,18].

In this study, we validated a protocol for the expansion of NPM1^mut^-specific cytotoxic T lymphocytes (CTLs) from both patients and healthy individual candidates donating hematopoietic stem cells for transplantation, and tested their potency and in vitro safety.

## 2. Materials and Methods

### 2.1. Patients and Samples

Bone marrow-derived mononuclear cells (BMMCs) from patients with *NPM1*-mutated AML at diagnosis or relapse and peripheral blood mononuclear cells (PBMCs) from the same patients at remission, or from healthy HSCT donors, were isolated by density gradient centrifugation, and cryopreserved [16]. Written informed consent was obtained from both patients and donors, according to the Declaration of Helsinki.

### 2.2. Production of Cytotoxic T Cell Lines Specific for NPM1^mut^

Dendritic cells (DCs) were obtained from PBMCs by CD14+ cell selection using CD14 Microbeads (Miltenyi Biotec GmbH, Bergisch Gladbach, Germany); positive selected cells were resuspended in supplemented medium adding interferon-α2b (Miltenyi Biotec GmbH), and granulocyte-monocyte colony-stimulating factor (GM-CSF, CellGenix, Freiburg, Germany) and cultured at 37 °C for 3 days.

CTL-NPM1^mut^ lines were obtained by culturing PBMCs from patients or donors with autologous dendritic cells pulsed with an NPM^mut^ peptide pool [17]. In detail, at day +5 of culture, autologous DCs were pulsed with the NPM^mut^ peptide pool at a final concentration of 5 µg/mL and irradiated (3000 rad). Donor or patient PBMCs were cocultured with DCs at a responder:stimulator ratio of 40:1 in RPMI 1640 medium supplemented with 5% autologous or healthy human serum pool and incubated at 37 °C in a humidified atmosphere at 5% CO_2_ for 7 days. At day +7–+9 and +14–+16, cultures were restimulated with a suspension of irradiated autologous feeder PBMCs (1 × 10^6^/well) pulsed with the NPM1^mut^ peptide pool, in the presence of 20 IU/mL recombinant human interleukin-2 (r-IL2) (Novartis, Basel, Switzerland), and 10 ng/mL recombinant human interleukin-15 (r-IL15, Miltenyi Biotec). On days +10–+12 and +17–+19, r-IL2 (20 IU/mL) and r-IL15 (10 ng/mL) were added to the cultures. At day +21/+23, the T cells obtained were collected, characterized for immune phenotype, and tested for potency in a standard ^51^Cr release cytotoxicity assay using P815 cell line, autologous or allogeneic phytohemagglutinin (PHA) blasts [16] pulsed or not with the NPM1^mut^ peptide pool or irrelevant peptides, and autologous or allogeneic AML blasts.

Furthermore, to test the product’s in vitro safety, growth inhibition of nonleukemic bone marrow-derived clonogenic progenitor cells derived from patients’ bone marrow cells at remission was also performed [19].

### 2.3. Immunophenotyping

NPM1^mut^-specific CTL products were characterized for phenotype by monoclonal antibody staining and flow cytometry. Anti-CD3 FITC, anti-CD4 PE, anti-HLA-DR PE, anti-CD8 APC, CDγδ FITC, anti-CD56 Pc5.5, anti-CD14 FITC, anti-CD56 PE, anti-CD3 Pc5.5, anti-CD19+ CD20 APC, anti-CCR7 FITC, and anti-CD45RA PE (Becton Dickinson, Franklin Lakes, NJ, USA) were employed.

### 2.4. NPM1-Specific T-Cytotoxic Activity Assessed as ^51^Cr Release

NPM1^mut^-specific T cells were tested for lytic activity towards different target cells, including autologous or allogeneic phytohemagglutinin (PHA) blasts [16] pulsed or not with the NPM1^mut^ peptide pool or irrelevant peptides, and autologous or allogeneic AML blasts. For the cytotoxicity assay, effector cells were incubated with target cells at effector/target (E:T) ratios from 20:1 to 0.01:1. Results are reported as percentage specific lysis at different E:T ratios [16]. The cytotoxic activity of the ATMPs was also assessed in a cytotoxicity assay against the NK-resistant cell line P815 in the presence and absence of anti-CD3 agonist antibody (OKT3), in order to assess the general lytic potential [17].

### 2.5. Cytokine Secretion Assessment by Enzyme-Linked Immunospot (ELISPOT) Assay or Flow Cytometry

The ability of NPM1^mut^-specific T cells to secrete a Th1 cytokine, i.e., interferon-γ (IFN-γ), was assessed in an ELISPOT assay [19]. A total of 1 × 10^5^ cells/well were stimulated for 20 h with the NPM1^mut^ peptide pool (final concentration of 50 μg/mL) [16]. Unstimulated ATMPs were used as negative controls. In detail, 96-well multiscreen filter plates (MAIPS 4510, Millipore, Bedford, MA, USA) were coated with 100 μL of primary antibody (IFN-γ, Mabtech, Nacka, Sweden) at 2.5 μg/mL, and incubated overnight at 4 °C. Cultured T cells were seeded in the absence or presence of the NPM1^mut^ peptide pool (final concentration of 50 μg/mL) [16]. After incubation for 24 h at 37 °C, 100 μL of biotinylated secondary antibody (Mabtech, 0.5 μg/mL) was added, and plates were then processed according to standard procedure. IFN-γ spots were counted using an ELISPOT reader (Bioline, Torino, Italy). The number of spots per well was calculated after subtracting assay background, defined as an average of the number of spots in 24 wells containing only complete medium, and specific background, defined as the number of spots in wells with responder alone.

Flow cytometry cytokine secretion assays for IFNγ, IL-2, and TNFα (Miltenyi Biotec) were also employed to functionally characterize the products, according to the manufacturer’s instructions. In detail, 10^6^ total cells were washed in cold buffer (phosphate buffered saline containing 0.5% bovine serum albumin and 2 mM EDTSA), centrifuged at 4–8 °C and after removing supernatant, were resuspended in RPMI 1640 medium (Life Technologies, Carlsbad, CA, USA) containing 5% human serum. IFNγ, IL-2, and TNFα Catch Reagent (Fitc, PE and APC conjugated, respectively) was added, and the sample was incubated on ice. After dilution with warm medium and incubations at 37 °C, and on ice, IFNγ, IL-2, and TNFα detection antibodies were added, together with anti-CD4 and -CD8 mAb. At the end of incubation, the cells were washed by cold buffer and analyzed by flow cytometry.

### 2.6. Clonogenic Assay

NPM1^mut^-specific T cells were incubated with marrow mononuclear cells in Iscove’s Modified Dulbecco’s Medium (IMDM, Gibco, Life Technologies) supplemented with 10% FBS at 37 °C in a 5% CO_2_ humidified atmosphere for 4 or 24 h. The cells were then collected, centrifuged, resuspended in 200 µL of IMDM and plated in a clonogenic assay for the growth of erythroid (BFU-E), and granulocytic-macrophagic (CFU-GM) colonies [20]. Cells were plated in 30 mm Petri dishes in 0.9% methylcellulose with 30% FCS, 10 ng/mL interleukin 3 (IL-3) (Miltenyi Biotec), 50 ng/mL granulocyte-macrophage colony-stimulating factor (GM-CSF) (CellGenix), and 3 IU of erythropoietin (rhEpo). After 14 days, colonies were counted using standard criteria.

### 2.7. Statistical Analysis

Data were described as the median and range if continuous and as count and percentage if categorical. To determine differences among patient groups, categorical variables were compared by chi-squared analysis, continuous variables with *t*-tests, and, if skewed, with non-parametric tests (Mann–Whitney U test); *p*-values < 0.05 were considered statistically significant. NCSS System (NCSS, Cary, NC, USA) was used for computation.

## 3. Results

### 3.1. NPM1^mut^-CTLs Can Be Expanded from AML Patients and Healthy Donors by Stimulation with NPM1^mut^-Peptide Pools

To evaluate whether NPM1^mut^-CTLs could be expanded from the peripheral blood of AML patients in remission, and potential stem cell donors, PBMC were stimulated or primed with autologous DC pulsed with the NPM1^mut^-peptide pool in the presence of IL2 and IL15. After one round of re-stimulation, T cells expanded from 3 patients and 7 donors showed a mean expansion of 11.1-fold (range 5.7–31).

The ATMP were tested for microbiological safety by sterility testing in an automated blood culture system, and by evaluation of endotoxin levels and mycoplasma contamination. The T cell products were found to be sterile, and with endotoxin levels within the acceptable range.

The recovered cells were predominantly CD3+ (mean 92%) with balanced predominance of CD4+ and CD8+ T cells (median 44%, range 26–63% and mean 48%, range 10–59%, respectively). CTLs contained variable low numbers of natural killer CD3-CD56+ cells (median 9%, range 2–20%) (Figure 1). Analysis of memory phenotype showed a slight prevalence of CD45RO+/CCR7− T cells in accordance with an effector memory phenotype, but central memory and naïve T cells were also present.

To demonstrate the presence of cytotoxic T cells within the product, we employed a CD3-redirected cytotoxicity assay. The T cell lines from both patients and donors mediated killing of P815 cell line induced by binding through OKT3 monoclonal antibody. The median percentage lysis at an effector to target (E:T) ratio of 5:1 was 52.5% (range 14–97%) (Figure 2).

We then proceeded to evaluate specific activity of the CTLs against the NPM1^mut^antigen. All T cell lines from patients and donors showed specific lysis against PHA blasts pulsed with the NPM1^mut^ peptide pool. In detail, we observed a median percentage lysis at the 5:1 effector to target (E:T) ratio of 35% (range 15–52), with median percentage lysis against control-pulsed PHA blasts of 3 (range 0–11) (Figure 2). To confirm the ability of NPM1^mut^-CTLs to recognize and kill leukemic cells, lytic activity against autologous (in the case of patients) or HLA partially-matched *NPM1*^mut^-positive leukemic blasts was also tested. Both patient-derived and donor-derived NPM1^mut^-CTLs mediated a measurable cytotoxic activity against their target population, with a median percentage lysis of 22% (range 10–59) at E:T ratio of 5:1 (Figure 2). We hypothesize that the activity we observed against allogeneic leukemia blasts in NPM1^mut^-CTLs from healthy donors was directed against leukemia, and not against alloantigens, as median cytotoxicity against the allogeneic non-pulsed PHA blast counterpart was 2% (range 0–9) (Figure 2 and Figure 3).

In order to verify that the response was directed to the mutated form of NPM1 antigen, rather than the wild-type form, we performed in vitro experiments using NPM1^mut^-CTLs from 3 donors, and testing cytotoxicity against PHA blasts pulsed with the NPM1^mut^ peptide pool and the HL-60 cell line that is positive for NPM1-wt. We observed a median percentage lysis at the 5:1 effector-to-target (E:T) ratio of 25% (range 14–49) for NPM1^mut^ peptide-pulsed targets, with median percentage lysis against HL-60 cells of 8 (range 0–12) (*p* < 0.05) (Figure 4). These results indicate a specificity of the CTL lines for the mutated form, although low lysis to NPM1wt antigen could be observed for some CTL lines.

We subsequently analyzed whether there was a difference in the specific response of patients’ and donors’ CTLs. We observed comparable leukemia-specific activity in the two cohorts, as median lysis at 5:1 E:T ratio against PHA blasts pulsed with the NPM1^mut^ peptide pool was 32% for donor CTLs compared with 38% for patients’ products (*p* = ns), and median cytotoxicity against autologous (in the case of patients) or HLA partially matched LB (donor CTLs) was 22 and 21%, respectively (*p* = ns). Only for CD3-redirected cytotoxicity, CTLs expanded from donors exhibited a higher, although not statistically significant, lytic potential than patients’ CTLs (median lysis of 74 vs. 41%, respectively, *p* = 0.11).

Consistent with the cytotoxicity results, NPM1^mut^-directed cytokine production, measured in a ELISpot assay, confirmed specificity of the CTLs. Antigen recognition in IFNγ-ELISpot showed a mean of 170 SFU/10^5^ cells (range 71–266) (Figure 5). The CTLs had a polyfunctional profile, as among CD8+ T cells, a median of 21% cells (range 2–30) were triple-positive for IFNγ, IL-2, and TNFα, while 15% CD4+ T cells (range 2–22) were triple-positive. Additionally in this case, we did not observe significant differences in the IFNγ-secreting activity of patients’ CTLs compared with donor T cell lines (median SFU/10^5^ cells 176 vs. 162, respectively).

### 3.2. NPM1^mut^-CTLs Show Limited Inhibition of CFU-GM, but Not of BFU-E, When Cocultured with BM Progenitors

NPM1 antigen is present in its non-mutated wild-type (WT) form in healthy myeloid precursors, and although peptides derived from the mutated protein have been used to stimulate CTL expansion, it is theoretically possible that specific NPM^mut^ CTLs could recognize peptides of WT protein on BM precursors, and that this mechanism leads to bone marrow suppression and toxicities. To evaluate the bone marrow suppressive effect of NPM1^mut^-CTLs in vitro, CTLs were cocultured with bone marrow samples from AML patients in remission, matched in at least one HLA antigen, for 4 or 24 h. As control conditions, BM cells were incubated with unmanipulated PBMCs from the same individuals.

At the end of coculture, cells were recovered, resuspended in the appropriate media, and plated in clonogenic assays for BFU-E and CFU-GM for 14 days. Colonies were then counted.

The results show that an inhibitory effect was observed mainly after 24 h incubation, and it concerned CFU-GM (median number of colonies, baseline: 20 vs. 13 after incubation with CTL, *p* = 0.35), while no effect was observed on BFU-E, which were increased in number after CTL co-incubation (median number of colonies, baseline: 14 vs. 20 after incubation with CTL, *p* = 0.24) (Figure 6).

Figure 7 shows the effect of CTL coculture time on BM progenitor colony growth. At 4 h, the inhibition on CFU-GM is less prominent.

## 4. Discussion

Despite remarkable advances in the treatment of *NPM1*-mutated AML, due to optimization of conventional induction chemotherapy and risk-stratified consolidation with cytarabine and HSCT in first remission in younger patients, and development of novel therapeutic approaches including BCL-2 inhibitor venetoclax and hypomethylating agent association, immune checkpoint inhibitors and AML-targeted monoclonal antibodies in older unfit patients, about 50% of patients still die of progressive disease [12,21,22]. Thus, there is a need for new therapies. Beyond novel agents such as menin inhibitors or the nucleolar stress triggerer dactinomycin [23,24,25,26], which have shown anti-leukemic activity in preclinical models and are being investigated in clinical trials, cell therapy approaches targeting NPM1-mutated protein on AML cells could represent a useful complementary approach to the other available treatment options. Our study shows the feasibility and reproducibility to expand NPM1^mut^-specific CTLs on a GMP scale from AML patients in hematologic remission, but also from healthy donors, by stimulation with DCs pulsed with a peptide pool derived from the mutated NPM1 antigen, irrespective of the donor HLA type. These CTLs, which include both CD8+ and CD4+ T cells, have high cytotoxic potential, as they are able to recognize and lyse target cells pulsed with NPM1^mut^ peptides and, more importantly, they efficiently kill autologous or, in the case of donor, HLA partially matched *NPM1*^mut^-positive leukemia blasts.

In the past, a number of studies have demonstrated a correlation between the emergence of tumor antigen-targeted T cells in the peripheral blood and/or bone marrow of leukemia patients after induction and maintenance chemotherapy, or after allogeneic HSCT, and long-term disease remission [17,27,28,29,30,31]. In patients with *NPM1*-mutated AML, a better overall survival was documented in the patients that developed autologous NPM1^mut^-specific T cell responses [15]. In line with this observation, our group was able to demonstrate, through sequential monitoring of IFNγ-producing NPM1^mut^-specific T cells coupled with molecular MRD [32,33,34], that the kinetics of leukemia-specific T cells inversely correlated with molecular or morphologic leukemia status, having detected increased and sustained specific immune responses in patients with persistent molecular CR, in some cases years after completion of leukemia treatments [16]. These data suggest the potential for an autologous NPM1^mut^-specific CTL product to promote leukemia control by eradicating persistent MRD or low-tumor burden relapse following conventional chemotherapy in older *NPM1*-mutated AML patients not eligible for allogeneic HSCT, as we observed in a patient treated for Ph+ALL with B-cell receptor-ABL (BCR-ABL) p190-directed CTLs [35].

The feasibility to expand leukemia-specific CTLs from healthy donors by leukemia antigen-derived peptide stimulation had been first shown in the setting of chronic myeloid leukemia (CML) by using the B-cell receptor-ABL (BCR-ABL) p210 fusion protein, proteinase 3 (Pr3) and Wilms’ tumor antigen 1 (WT1) antigens [36], and subsequently replicated for BCR-ABL p190 [35], WT1 alone [37,38] or combined with multiple antigens including Pr3, human neutrophil elastase (NE), melanoma-associated antigen A3 (MAGE-A3), preferentially expressed antigen in melanoma (PRAME) and survivin [11,18,39]. In the case of *NPM1*-mutated AML, we were successful in priming leukemia-specific CTLs able to kill efficiently partially HLA-matched primary myeloid leukemia blasts from all donors tested. This finding reflects previous observations from our and other groups on the high frequency of IFNγ-producing NPM1^mut^-specific T cells in the peripheral blood of healthy volunteers or AML patients after HSCT [16,40], possibly due to a cross-reactive immune response induced by short amino acid sequences from the C-terminal of *NPM1*-mutated protein homologous with several common viral and bacterial antigens [16]. Successful expansion of CTLs from individuals with varied HLA types was obtained by employing a pool of fifteen 9- and 11-mer NPM1^mut^ peptides, but also three 18-mer peptides that could contain multiple T cell epitopes, possibly cross-presented through various HLA class I and II alleles [41]. In addition, the peptides included epitopes representative of the most common *NPM1* gene mutations, in order to allow broad antigen targeting [16].

The use of a donor-derived NPM1^mut^-specific product could be usefully employed as targeted DLI in MRD-positive patients to prevent hematologic relapse after allogeneic HSCT. Indeed, some of these products were infused prophylactically [18,35,39] or pre-emptively [18,36] in patients with CML or acute leukemia who were at high risk of relapse or had positive MRD after allogeneic HSCT, demonstrating safety and in vivo antileukemia effects. Although the design of these early phase trials did not allow assessing with certainty the role of leukemia antigen-targeted DLIs in patient outcomes, as patients generally received associated therapies, it is undeniable that these cohorts had at least comparable efficacy to chemotherapy and unmanipulated DLI, with a reduced risk of GVHD as compared with DLI [18,39].

Although a preemptive strategy guided by MRD monitoring could, more successfully, guarantee leukemia control due to the low tumor burden, a curative treatment approach in patients with hematologic relapse after HSCT may also be pursued. Indeed, in some of the early phase studies, responses were registered in patients with active disease [35,39]. In this latter setting, it is unlikely that the use of cell therapy alone, as observed for hypomethylating agent (HMA)-based salvage therapy [42,43,44], will be sufficient to control the outgrowth of leukemia blasts long-term, except perhaps in a few patients with late relapse and low tumor burden [44]. However, the use of targeted cell therapies that include a higher number of leukemia-specific T cells compared with unmanipulated DLIs, combined with azacytidine and/or the BCL-2 inhibitor venetoclax, or FLT3 inhibitors such as gilteritinib or sorafenib [45], may provide an added advantage and be able to increase the response rate in patients with hematologic relapse after HSCT. Indeed, HMA have been shown to induce HLA class I and costimulatory molecule expression on leukemia blasts, and favor their susceptibility to T cell-mediated cytotoxicity, while venetoclax was able to directly enhance cytotoxicity against AML cells both in vitro and in vivo [46,47].

So far, no specific T cell therapy for *NPM1*-mutated AML has been employed in vivo. Van der Lee et al. have been able to clone and transfer a *NPM1*-mutated, HLA-A2-restricted, T cell receptor with efficient in vitro specificity against NPM1-mutated primary leukemic blasts and in vivo activity in immunodeficient mice engrafted with a human *NPM1*-mutated AML cell line [10]. The advantages of gene therapy are the relatively short time required for T cell production, and the possibility to introduce the TCR into different T cell subsets with higher in vivo persistence and antitumor efficacy, such as central memory or stem cell memory T cells. On the other hand, a gene therapy product could induce more severe toxicity than somatic cell therapy [48], and be difficult to modulate in older, unfit patients, or in the early post-transplant setting.

One other problem of ATMP therapy, which has emerged in the past few years in connection with CAR-T cell therapy, is the development of tumor immune escape by several mechanisms, including loss of antigen expression [49]. In the case of NPM1^mut^-specific products, immune escape is more unlikely, as *NPM1* mutation is a driver of genetic lesion, critical for leukemia cell survival. The use of associated therapies could provide the means to further broaden the response to leukemia, and avoid development of immune escape from antigen loss, by lysing leukemia cells in vivo and inducing antigen spreading [35]. This, in turn, could stimulate either recruited endogenous T cells emerging from the graft, or some T cells with low NPM1^mut^ fitness present in the bulk ATMP product. Stimulation with long peptides presented by DCs allowed for expansion of CD4+ T cell populations alongside CD8+ CTLs. CD4+ T cells in the products, in addition to providing T cell help to CD8+ CTLs, were also endowed with lytic activity, as we could observe cytotoxicity mediated by this subset. This characteristic of the NPM1^mut^-specific CTL product could be of advantage to counteract immune escape mechanisms developed by leukemia cells, in particular downregulation of surface HLA class I molecules, necessary for CD8+ CTL activity. Virus-specific CD4+ CTLs have been proven to play a protective role in antiviral immunity, when pathogens such as herpesviruses escaped from CD8-mediated cellular immunity by downregulating the expression of HLA class I on the surface of infected cells through inhibition of the TAP transporter and/or proteasome degradation pathways [50]. A similar mechanism has been suggested also in the setting of tumor immune surveillance [51,52].

The wild-type form of NPM1 (wt-NPM1) is expressed on all cells, including BM-resident CD34+ hematopoietic stem cells. Although the mutated form is a neoantigen almost exclusively found in AML [14], and generally expressed in the entire leukemic population, while not detectable in clonal hematopoiesis, there is a theoretical concern that NPM^mut^ CTLs, once administered to patients, could recognize wt-NPM1 on non-leukemic hematopoietic progenitors and mediate BM toxicity. To analyze this potential cross-reactivity, inhibition assays of BM precursor colony formation were performed by co-incubation of patients’ bone marrow cells with scaled ratios of NPM^mut^ CTLs, and subsequent culture. The results of clonogenic assays showed that NPM^mut^- CTLs did not significantly affect the growth of patient-derived normal progenitor cells. These data are in line with a recent report that described a high response to wt-NPM1 by CD8+ CTLs in an NPM1-mutated AML patient post-HSCT [53]. In this specific patient, the emergence of a cellular immune response to wt-NPM1 was not accompanied by side effects, such as GVHD or BM aplasia, and coincided with leukemia control, further suggesting the potential role and relative safety of NPM1-directed immunity after HSCT.

## 5. Conclusions

Patients with relapsed or refractory *NPM1*-mutated AML after allogeneic HSCT or unfit patients relapsing after chemotherapy are an unmet medical need. Novel targeted therapies are emerging, and may change the prognosis of these cohorts. However, additional, complementary strategies, such as mutated-NPM1 specific cellular therapy, could represent a tool to prevent tumor outgrowth and escape, and further increase the probability of leukemia control.

Our findings indicate that NPM1^mut^-CTLs, obtained from patients or healthy donors and endowed with leukemia-specific activity, may constitute a safe and effective option in the treatment of patients affected by refractory or relapsed *NPM1*-mutated AML.

## Figures and Tables

**Figure 1 cancers-15-02731-f001:**
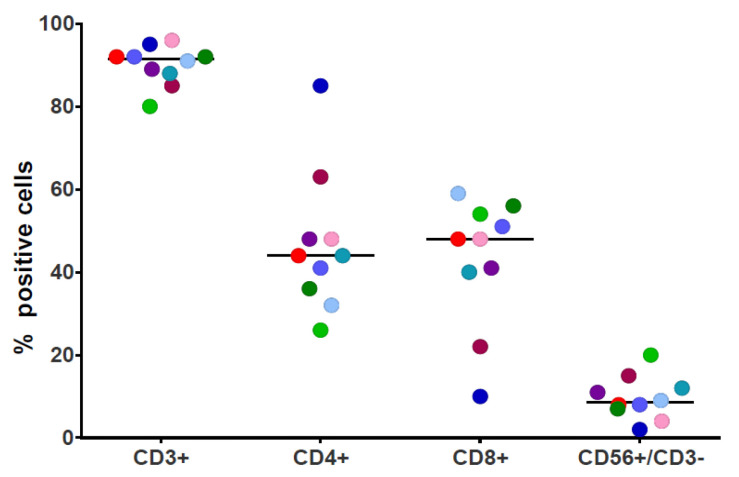
Surface phenotype of NPM1^mut^-CTLs. The results are reported as % positive cells. The 3 patients are indicated with light green, light blue, and red dots.

**Figure 2 cancers-15-02731-f002:**
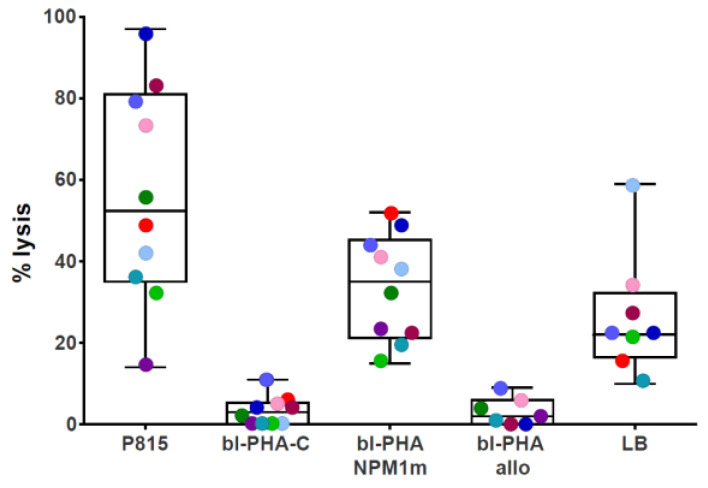
Potency of NPM1^mut^-CTLs: cytotoxic activity. The lytic activity of NPM1^mut^-CTLs against P815 cell line (CD3-redirected cytotoxicity), against control (bl-PHA-C) or NPM1^mut^ peptide pool-pulsed autologous PHA blasts (bl-PHA-NPM1m), against allogeneic PHA blasts (bl-PHA-allo), and against autologous or allogeneic leukemia blasts (LB) is shown. The results are reported as % lysis at an E:T ratio of 5:1. The 3 patients are indicated with light green, light blue, and red dots.

**Figure 3 cancers-15-02731-f003:**
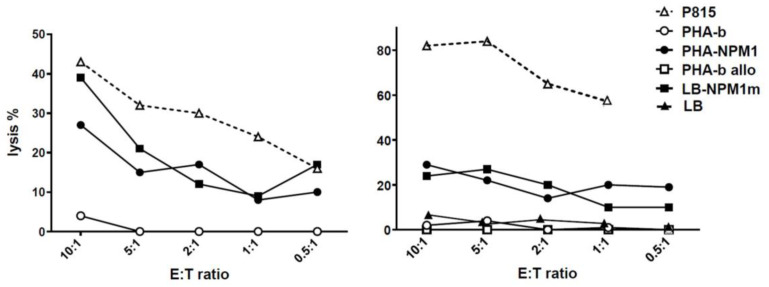
Potency of NPM1^mut^-CTLs: cytotoxic activity. Exemplificative cytotoxicity experiments of NPM1^mut^-CTLs from an AML patient (**left**) panel and a healthy donor (**right**) panel are shown. Lysis against P815 cell line (CD3-redirected cytotoxicity, white triangles), against control (PHA-b, white circles) or NPM1^mut^ peptide pool-pulsed autologous PHA blasts (PHA-NPM1, black circles), against allogeneic PHA blasts (PHA-b allo, white squares), and against autologous or allogeneic NPM1m-positive leukemia blasts (LB-NPM1m, black squares) or allogeneic NPM1m-negative LB (LB, black triangles), is reported as % lysis at different E:T ratios.

**Figure 4 cancers-15-02731-f004:**
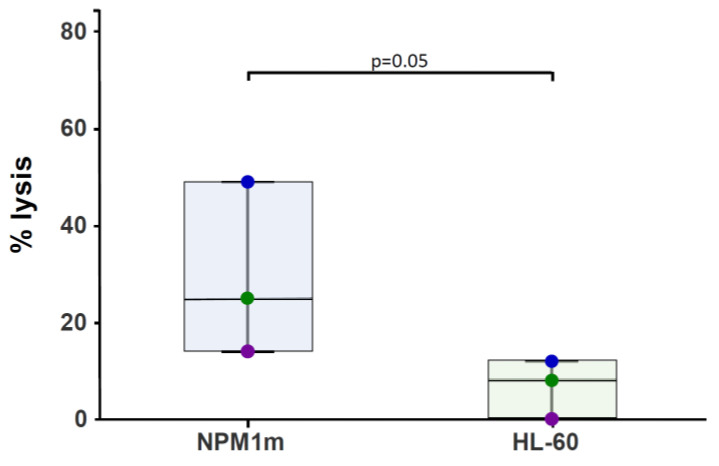
Potency of NPM1^mut^-CTLs: cytotoxic activity. The lytic activity of NPM1^mut^-CTLs against NPM1^mut^ peptide pool-pulsed autologous PHA blasts (bl-PHA-NPM1m), and against the *NPM1*wt-positive HL-60 cell line is shown. The results are reported as % lysis at a E:T ratio of 5:1.

**Figure 5 cancers-15-02731-f005:**
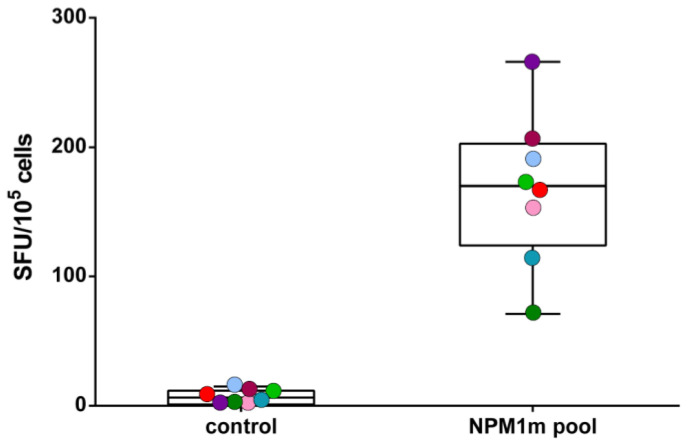
Potency of NPM1^mut^-CTLs: IFNγ secreting activity. IFNγ production by NPM1^mut^-CTLs in the absence of specific stimulation (control) or after stimulation with NPM1^mut^ peptide pool (NPM1m pool) is shown. The results are reported as spot-forming units (SFU)/10^5^ CTLs. The 3 patients are indicated with light green, light blue, and red dots.

**Figure 6 cancers-15-02731-f006:**
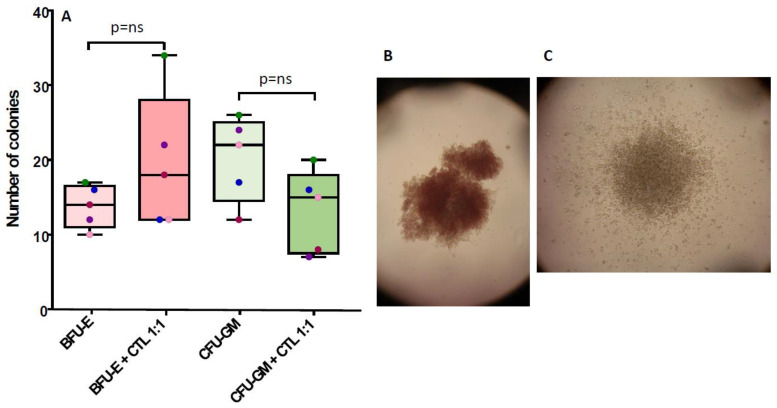
In vitro safety of NPM1^mut^-CTLs: effects of CTL co-incubation on BM progenitor cells. (**A**) The number of BFU-E (pink boxes), and CFU-GM (green boxes), at baseline (lighter color boxes) or after 24 h co-incubation with CTLs (darker boxes) are shown. In (**B**,**C**), examples of BFU-E and CFU-GM are shown.

**Figure 7 cancers-15-02731-f007:**
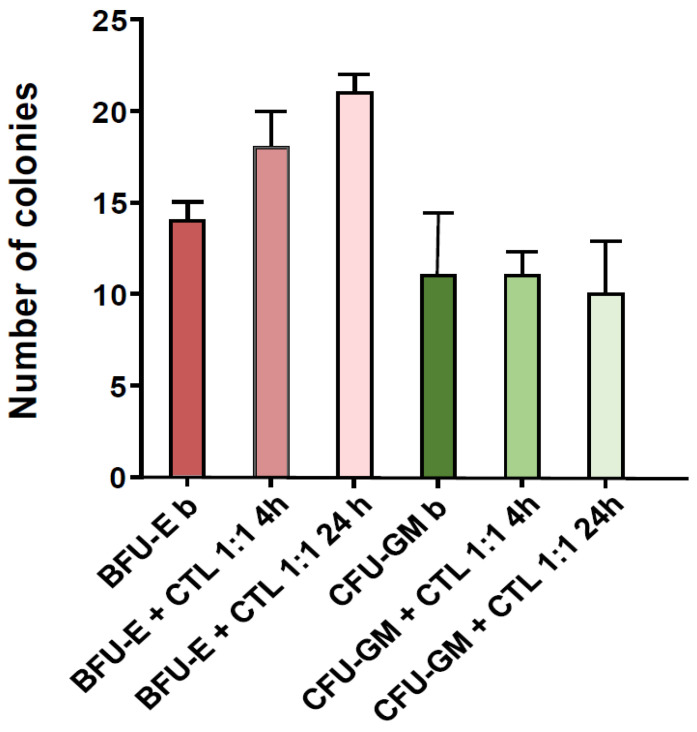
In vitro safety of NPM1^mut^-CTLs: effects of CTL co-incubation duration on BM progenitor cells. The number of BFU-E (pink columns), CFU-GM (green columns) at baseline (darker color column), or after 4 h (medium color column) and 24 h (lighter color column) co-incubation with CTLs, are shown for an exemplificative CTL line.

## Data Availability

The clinico-laboratory data of this study are available on reasonable request from the corresponding authors, according to privacy and ethical restrictions.

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
