# Peer review of "Preclinical Validation of an Advanced Therapy Medicinal Product Based on Cytotoxic T Lymphocytes Specific for Mutated Nucleophosmin (NPM1mut) for the Treatment of NPM1mut-Acute Myeloid Leukemia"

_cancers, 2023, doi:10.3390/cancers15102731_

Round 1

Reviewer 1 Report

De Cicco and colleagues have analyzed the feasibility and reproducibility of a method to expand NPM1mut-specific CTLs from AML patients in hematologic remission and healthy controls. The method is based on the stimulation of lymphocytes by  DCs pulsed with an NPM1-mutated peptide. The authors show that NPM1mut-CTLs can be expanded from AML patients and healthy donors and that these cells exhibited cytotoxic activity and secrete IFN gamma. Overall, the manuscript is well-written and the experimental design adequate, but a few concerns must be addressed to support the authors' conclusions.

The main point is to confirm the specificity of the antileukemic effect. Considering that several indirect mechanisms may have been contributing to cytotoxicity,  the authors need to show that the target is indeed the mutated NPM1 protein on the surface of cells. 

Author Response

We thank the reviewer for the kind words of appreciation for our preliminary work.

  1. The main point is to confirm the specificity of the antileukemic effect. Considering that several indirect mechanisms may have been contributing to cytotoxicity,  the authors need to show that the target is indeed the mutated NPM1 protein on the surface of cells.

As suggested by the reviewer, we performed experiments to demonstrate that NPM1mut-specific CTLs mediate low killing activity towards AML-derived cell lines known to be NPM1wt carriers, such as HL-60. Data have been inserted in the text (page 6, lines 258-265) and new Figure 4. 

Reviewer 2 Report

The study demonstrates how T cells specific for mutated antigen can be expanded and able to recognize and lyse leukemia blasts carrying NPM1 mutated antigen in a safe way for other blood cells. This work can have a potential impact on the actual therapy of some form of Leukemia and for this reason this study represents a big novelty in this field.

The paper is sufficiently well-written as well the experiments, although few, are well-perfomed and cleary presented.

Here, some minor requestions:

-Though the data are represented in a clear way, in each Figure, the standard deviation and significativity should be shown.

-In clonogenic assay, images of colonies should be reported.

-Some experiments in which specific T cells versus NPM1 mutated blasts and chemotherapies, usually used in clinic therapy, should be performed in order to demonstrate the existence of a synthetic letality of this novel potential therapy.

Author Response

We thank the reviewer for the kind words of appreciation for our preliminary work.

Answers to queries:

  1. Though the data are represented in a clear way, in each Figure, the standard deviation and significativity should be shown.

Some of the Figures already report median and 95% confidence interval. We have further modified the Figures to include significance and SD, when not previously reported.

  1. In clonogenic assay, images of colonies should be reported.

We inserted images of colonies in Figure 6.

  1. Some experiments in which specific T cells versusNPM1 mutated blasts and chemotherapies, usually used in clinic therapy, should be performed in order to demonstrate the existence of a synthetic letality of this novel potential therapy.

Cytotoxicity of specific T cells versus NPM1 mutated blasts were performed, and are shown in Figures 2 and 3. We did not include CTLs pre-treatment with chemotherapy, as the T cells from the patients were already obtained after administration of chemotherapy (so, partly pre-treated); moreover, previous work from our group demonstrated the ability of NPM1-mutated AML patients to mount in vivo a specific response while receiving treatment (Forghieri et al, Oncotarget 2019), and BCR-ABL-specific CTLs obtained with a protocol similar to the one described in this paper were successfully employed in patients receiving TK inhibitors for BCR-ABl-positive ALL (Comoli et al, Blood 2017).   

Reviewer 3 Report

De Cicco et al provide the first biological data regarding the feasibility of producing mutated-NPM1-specific cytotoxic T cells (CTLs) suitable for somatic cell therapy to prevent or treat hematologic relapse in patients with NPM1-mutated AML.

With a precise and well detailed methodology, the Authors were able to expand in vitro CTLs from both pts and healthy donors able to induce a cytotoxic effect directed and limited to NPM1 mutated cells, saving the normal HSC. 

Moreover, even if the source is different (i.e., pts in remission or healthy donors), it seems there are no differences in the efficacy of blast-killing activity. This something interesting, considering that this can give us the possibility to have an “off the shelf” cellular product, which does not need any cellular engineering. 

I have only one minor comments for the Authors: did you observe any difference in the cytoxicity or in the amount of successfully activated cells between patients in remission and healthy donors? 

Author Response

  1. I have only one minor comments for the Authors: did you observe any difference in the cytoxicity or in the amount of successfully activated cells between patients in remission and healthy donors? 

We thank the reviewer for the kind words of appreciation for our preliminary work.

To answer the question, we did not observe major differences in the activity shown by the CTL lines obtained from donors or patients, with the caveat of the small number of experiments performed in the AML patient cohort. As pointed out in the results section, we observed comparable leukemia–specific activity in the two cohorts, as median lysis at 5:1 E:T ratio against PHA blasts pulsed with the NPM1mutpeptide pool was 32% for donor CTLs compared with 38% for patients’ products (p=0.93 by Mann Whitney test), and median cytotoxicity against autologous (in the case of patients) or HLA-partially matched LB (donor CTLs) was 22% and 21%, respectively (p=0.67). Only for CD3-redirected cytotoxicity, CTLs expanded from donors exhibited a higher, although not statistically significant, lytic potential than patients’ CTLs (median lysis of 74% vs 41%, respectively, p=0.11). On the contrary, expansion of NPM1-specific T cells was higher, although not statistically significant, in patients than in donors  (median 18-fold in patients vs 8 fold in donors, p=0.17).

Reviewer 4 Report

In their paper De Cicco et al addressed the development of a cellular therapy specific for NPM1-mutated AML and based on cytotoxic T lymphocytes. They describe a protocol for the expansion of T lymphocytes from both patients and healthy individuals. The methods are well described and extremely detailed. The overall approach deserves merit.

Major issues

The introduction on the subject of the clinical management of patients diagnosed with NPM1-mut AML should be integrated, as in this specific setting low intensity approaches as those including venetoclax and hypomethylating agents has been proven to be effective and tolerable. The authors should modify their background positioning their proposal of cellular therapy accordingly. The discussion is well balanced and provides a detailed description of the current scenario of cellular therapy especially in the after HSCT setting, and also deepens some technical issues related to this therapeutic approach.

Minor issues

The names of the genes (i.e., NPM1) should be reported in italics.

Line 285: “1” is missing from NPM1

Author Response

Major issues

  1. The introduction on the subject of the clinical management of patients diagnosed with NPM1-mut AML should be integrated, as in this specific setting low intensity approaches as those including venetoclax and hypomethylating agents has been proven to be effective and tolerable. The authors should modify their background positioning their proposal of cellular therapy accordingly. The discussion is well balanced and provides a detailed description of the current scenario of cellular therapy especially in the after HSCT setting, and also deepens some technical issues related to this therapeutic approach.

We thank the reviewer for the favorable comments. 

As suggested, we have modified the introduction by integrating comments on the positioning of cell therapy in the setting of low intensity approaches (page 2, lines 64-66, 70-71, 95). 

Minor issues

  1. The names of the genes (i.e., NPM1) should be reported in italics.

 We have corrected the fonts for gene names.

  1. Line 285: “1” is missing from NPM1

We apologize for having overlooked the mistake, that has now been corrected.

Round 2

Reviewer 1 Report

The authors have included samples of HL-60 cells, which are NPM1 wild-type, in their experiments. The demonstration of higher cytotoxic efficiency of CTLs against the NPM1-mutant cells reinforces the authors' claims about specificity.